# Direct dehydrocoupling facilitates efficient thiophene anchoring on silicon surfaces

Jingpeng Li, Meiyu Zhang, Wenxuan Li, Zhongshu Li [ID], Tingshun Zhu [ID] & Zhenyu Yang [ID] ✉

Silicon is a cornerstone material in electronics and photovoltaics due to its abundance, tunable semiconducting properties, and chemical versatility. Direct anchoring of thiophenes, with their highly delocalized aromatic backbones, onto silicon surfaces offers a promising route to tailor charge carrier migration properties. However, current methods for anchoring thiophenes commonly rely on pre-activation of precursors or transition-metal catalysts. Here, we introduce a catalyst-free radical strategy for direct linkage of thiophenes with Si atoms on organosilanes and silicon surfaces. This method leverages thermally induced homolytic cleavage of Si-H bonds to generate silicon radicals, which undergo efficient hydrosilylation with thiophene rings, forming Si-C linkages and releasing $H_2$. We demonstrate the successful application of this approach on silicon surfaces, achieving functionalization with thiophenes that enhance charge carrier mobilities in silicon nanocrystals significantly higher than previously reported alkyl-functionalized SiNCs, indicating the significant potential of catalyst-free dehydrocoupling for advancing silicon-based materials in optoelectronic applications.

Silicon is a fundamental element in the semiconductor industry due to its exceptional physical properties and chemical stability[1-3]. Its extraordinary semiconductor characteristics have revolutionized the electronics industry, establishing it as the foundational material for integrated circuits, transistors, and optoelectronic devices such as solar cells, light-emitting diodes, and photodetectors[4-6].

The versatility of silicon extends to surface chemistry, where its ability to form covalent bonds with various elements enables the modification of the surface silicon atoms on silicon-containing solids and even the construction of new silicon-based structures and hybrid materials[7-10]. This is typically achieved by replacing chemically active bonds on silicon surfaces (e.g., Si-H, Si-X, where X = Cl, Br) with more robust alternatives (e.g., Si-O, Si-C, and Si-N) through wet chemistry approaches such as hydrosilylation, hydroxylation, and dehydrocoupling reactions[11-13]. Among these methods, direct arylation of silicon surfaces has emerged as a promising strategy, allowing the covalent attachment of aromatic compounds without pre-functionalization or harsh reaction conditions[14]. The incorporation of aryl-based ligands enhances electronic coupling between silicon crystals and the frontier orbitals of the conjugating surface groups, providing additional flexibility to tailor the photophysical and electronic properties of silicon materials[15,16].

The introduction of thiophene-based ligands onto silicon surfaces represents a significant advancement in modulating charge carrier transportation of the covalent crystal[17]. Thiophene derivatives, with their delocalized aromatic carbon backbones, offer abundant near band-edge states, thereby enhancing interfacial charge carrier mobilities[18,19]. These structures can also improve spontaneous emission and modulate emission bands through extended π-conjugation effects via surface-ligand interactions[17,20].

However, current methodologies for directly anchoring thiophene species to silicon surfaces face challenges related to efficiency, scalability, and the stringent reaction conditions. The bridging between the backbone of thiophene species and silicon atoms commonly relies on dehydrogenative coupling strategies (Fig. 1), which typically involve the activation of carbon atoms on thiophene and the formation of highly

MOE Laboratory of Bioinorganic and Synthetic Chemistry, Lehn Institute of Functional Materials, School of Chemistry, IGCME, Sun Yat-sen University, Guangzhou, Guangdong, China. ✉e-mail: yangzhy63@mail.sysu.edu.cn

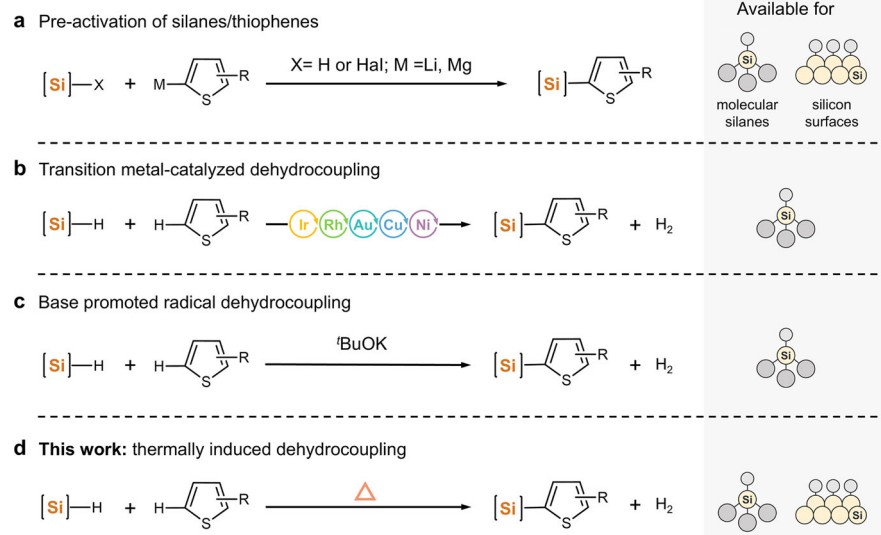

**Fig. 1 | Schematics of the existent dehydrogenative coupling methods between thiophene species and Si-H bonds. a** Including Grignard reactions between organometallic complexes and halide (Hal)-capped silicon. **b** Dehydrogenative coupling reactions catalyzed by transition metals or metal complexes. **c** Direct dehydrocoupling by using potassium ᵗBuOK induced Si-H cleavage to generate Si radicals. **d** Proposing that free radicals generated by thermally driven silicon undergo dehydrogenative coupling with thiophene species, without catalysts or radical initiators.

reactive silicon-based bonds (e.g., Si-H, Si-Cl, Si-Br)[11–13,17,21]. These methods often require pre-activation of Si-H and C-H bonds, which reduces manipulability and yield while introducing new leaving groups and by-products. Although transitional-metal-based catalysts can selectively activate C-H/Si-H bonds at relatively low temperatures, they may suffer from site selectivity issues and generated undesired by-products that requires additional purification processes[22–29]. Furthermore, residual ions or atoms can significantly impair the photophysical properties of crystalline silicon[22,30,31].

Differing to the existing transitional-metal-based catalysts, potassium tert-butoxide (ᵗBuOK) is a strong non-nucleophilic base that has recently been shown to promote efficient dehydrogenative coupling of molecular silanes and thiophenes under mild conditions[32,33]. This reaction relies on the generation of silyl radicals through base-promoted Si-H cleavage, which interact with thiophene heterocycles via an addition-restoration process to form Si-C linkages. While ᵗBuOK facilities reactions with various molecular silanes, its strong basicity poses a significant limitation for silicon surface applications, as it oxidizes hydride-terminated silicon surfaces rather than promoting dehydrocoupling (see Supplementary Note 1 and Supplementary Fig. 1 for details). A catalyst-free, direct method for dehydrocoupling thiophenes with Si-H on either molecular silanes or silicon surfaces remains elusive.

In this study, we propose that the formation of silyl radical (Si·) is the key step in promoting dehydrocoupling between C-H on thiophenes and Si-H. Inspired by the homolytic cleavage of Si-H bonds on silicon surfaces[34], we developed a direct dehydrocoupling strategy that thermally activates Si-H bonds without the presence of functional groups or any catalyst. This facilitates highly efficient, one-step thiophene functionalization on both molecular silanes and silicon surfaces. Without the interference of bases or oxidants, the thermally-activated Si· radicals directly interact with thiophene rings, followed by β-hydrogen cleavage to restore aromaticity and form Si-C bonds. We demonstrate that this method is applicable to silicon crystals of various sizes, significantly improving the stability and solution processability of silicon nanocrystals (SiNCs) while tuning their photoluminescent and optoelectronic properties. These thiophene-functionalized SiNCs exhibit enhanced hole and electron mobilities, reaching $1.52 \times 10^{-5}$ and $6.47 \times 10^{-6}\,cm^2 \cdot V^{-1} \cdot S^{-1}$, respectively, significantly higher than reported alkyl-passivated SiNCs[35–37].

## Results and discussion

### Dehydrocoupling reactions on organosilane

Inspired by the thermal generation of silicon radicals and their potential for dehydrogenative coupling with thiophene species, we developed a method that relies solely on heat to drive the Si· formation and facilitate their coupling with thiophene derivatives. We first investigated the effectiveness of the thermally driven Si-H cleavage in a homogenous reaction system. Triphenylsilane (TPS) served as the organosilane model, and benzothiophene (BT) was chosen as a representative thiophene species due to their suitable molecular weights and boiling points for the dehydrocoupling reaction at elevated temperatures. The reaction between TPS and BT was conducted at 180 °C to enable effective homolytic leavage of the Si-H bond and subsequent interaction with the C-H bond on the thiophene ring. After 12 h, the reaction mixture became cloudy, yielding the desired aromatic silane product BT-TPS with a 73% yield (Fig. 2a, measured by the weight of purified product after column chromatography). Proton nuclear magnetic resonance (¹H NMR) spectroscopy revealed a singlet proton peak at 7.55 ppm, confirming the successful substitution of the C-H bond at the C2 position of BT (see Supplementary Figs. 2 and 3), consistent with previous reports[38]. The regioselectivity was further verified by single-crystal X-ray diffraction (Fig. 2b and Supplementary Table 1).

To explore the generality of the reaction, we extended the thermally driven dehydrocoupling on TPS with two additional thiophene species: 2-hexylthiophene (1T) and oligothiophene 5-hexyl-2,2'-bithiophene (2T), under the same reaction conditions. NMR analysis (Fig. 2a) confirmed the formation of dehydrocoupled products with yields of 69% and 63%, respectively (Supplementary Figs. 4–7). The substitution of hydrogen at the C2 position of thiophenes was observed in the ¹H NMR spectra, indicating the high regioselectivity of the reaction.

### Mechanistic studies

Next, we conducted a kinetic study of the dehydrocoupling process using the TPS/BT reaction model, monitoring the product yield via ¹H NMR (Fig. 2c). Consistent with the generally accepted mechanism of Si-H bond homolytic cleavage, temperature plays a critical role in thermal hydrosilylation, as a critical temperature of 150 °C is required to

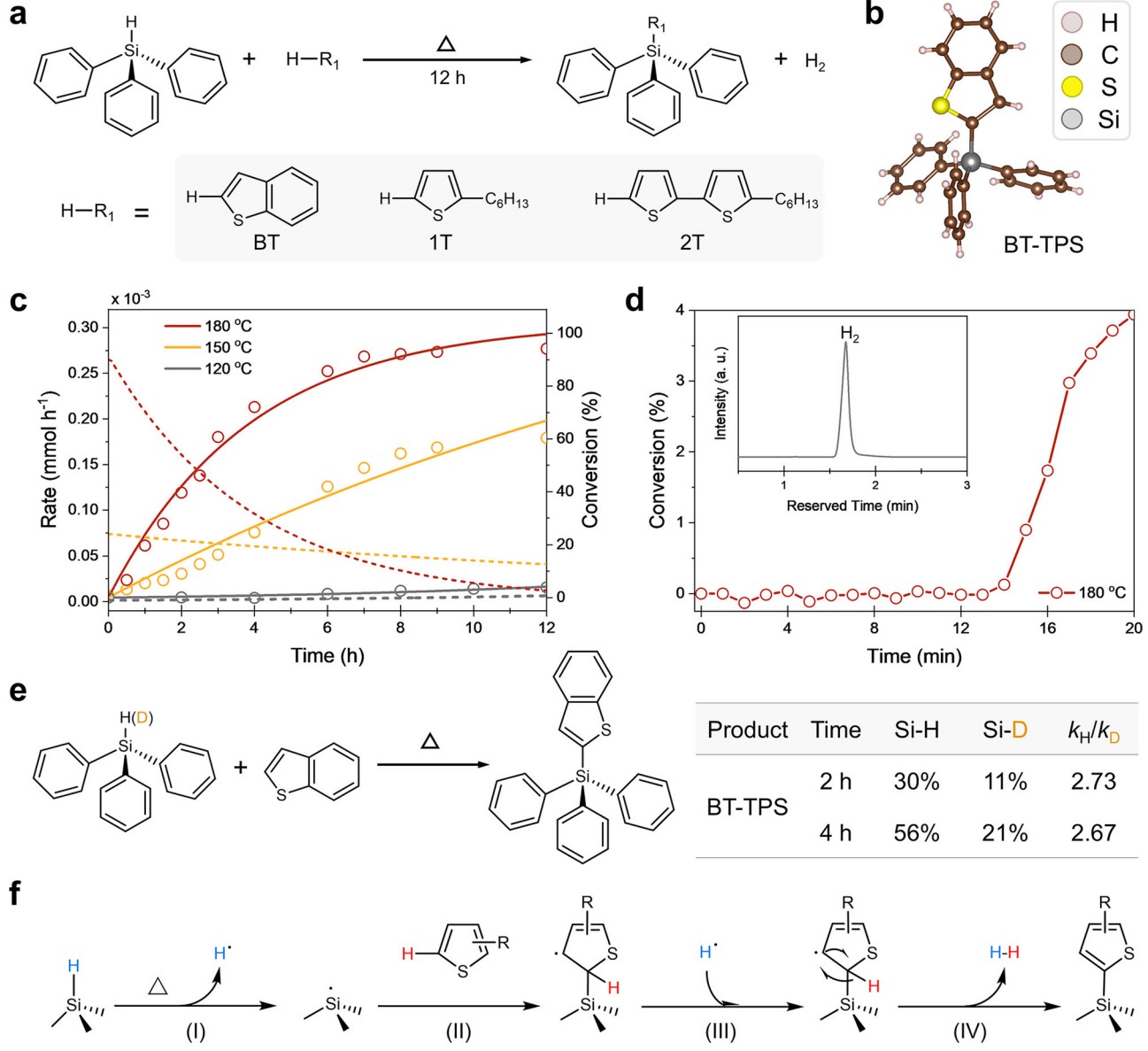

**Fig. 2 | Thermally induced dehydrocoupling reaction between molecular silane and thiophenes. a** General reaction pathway using TPS as the silane model. **b** Single crystal structure of the product BT-TPS (CCDC#: 2361418). **c** Kinetic study of the dehydrocoupling reaction between BT and TPS at various temperatures. Conversion values were calculated based on Si-H signal attenuation in $^1$H NMR spectra. **d** Representative time course of the dehydrogenative coupling of BT-TPS, monitored by in situ $^1$H NMR. **e** Investigation on the kinetic isotopic effects (KIEs) of thermally driven dehydrogenative coupling process. **f** Proposed reaction mechanism of the catalyst-free dehydrocoupling process between silanes and thiophenes, in which four key steps are included: (I) thermally induced homolytic cleavage of Si-H, (II) hydrosilylation with thiophenes, (III) aromatization and (IV) dehydrocoupling.

overcome the bond dissociation energy (BDE)[39]. This temperature dependence was also observed in our reactions: at 120 °C, almost no bubbles was observed, and only ~2% of the precursor reacted after 12 h. However, when the temperature was increased to ≥150 °C (regardless of ambient light), the reaction rate increased significantly to 0.050 mmol h$^{-1}$ (150 °C) and 0.083 mmol h$^{-1}$ (180 °C), comparable to the dehydrocoupling reactions promoted by transition metals or base metal (Supplementary Table 2). Notably, an induction period of ~20 min was observed at the beginning of the reaction at 180 °C (Fig. 2d), consistent with the behavior of dehydrocoupling reactions initiated by in situ radical formation[33]. It is important to note that although the temperature differs from those of $^t$BuOK-driven dehydrocoupling, the high reaction rate underscores the efficiency of this catalyst-free approach for direct thiophene-anchoring, which holds promise for scalable synthesis of silane derivatives and semiconductor surface treatment applications.

All these experimental results suggest that the dehydroucpling reaction is promoted by Si·, which are formed in situ through thermally driven homolytic cleavage of Si-H bonds. To verify this hypothesis, we employed 2,2,6,6-tetramethyl-1-piperidinyloxy (TEMPO), a well-known radical scavenger in silicon chemistry[40], to capture any Si· generated during the reaction. The addition of TEMPO to the mixture of TPS and BT significantly inhibited the reaction, and the TEMPO-incorporated silanol structure (TEMPO-TPS) was detected via mass spectroscopy (Supplementary Note 2 and Supplementary Fig. 8), confirming the presence of Si· as the key species promoting the dehydrocoupling process.

To further investigate the reaction kinetics, we performed parallel reactions using TPS and deuterated TPS (TPS(D)) with BT to compare their kinetic isotopic effect (KIE) values (Scheme 2e). The reactions were carried out for predetermined durations (2 h and 4 h), and then the dehydrocoupled product (i.e., BT-TPS) was isolated for yield

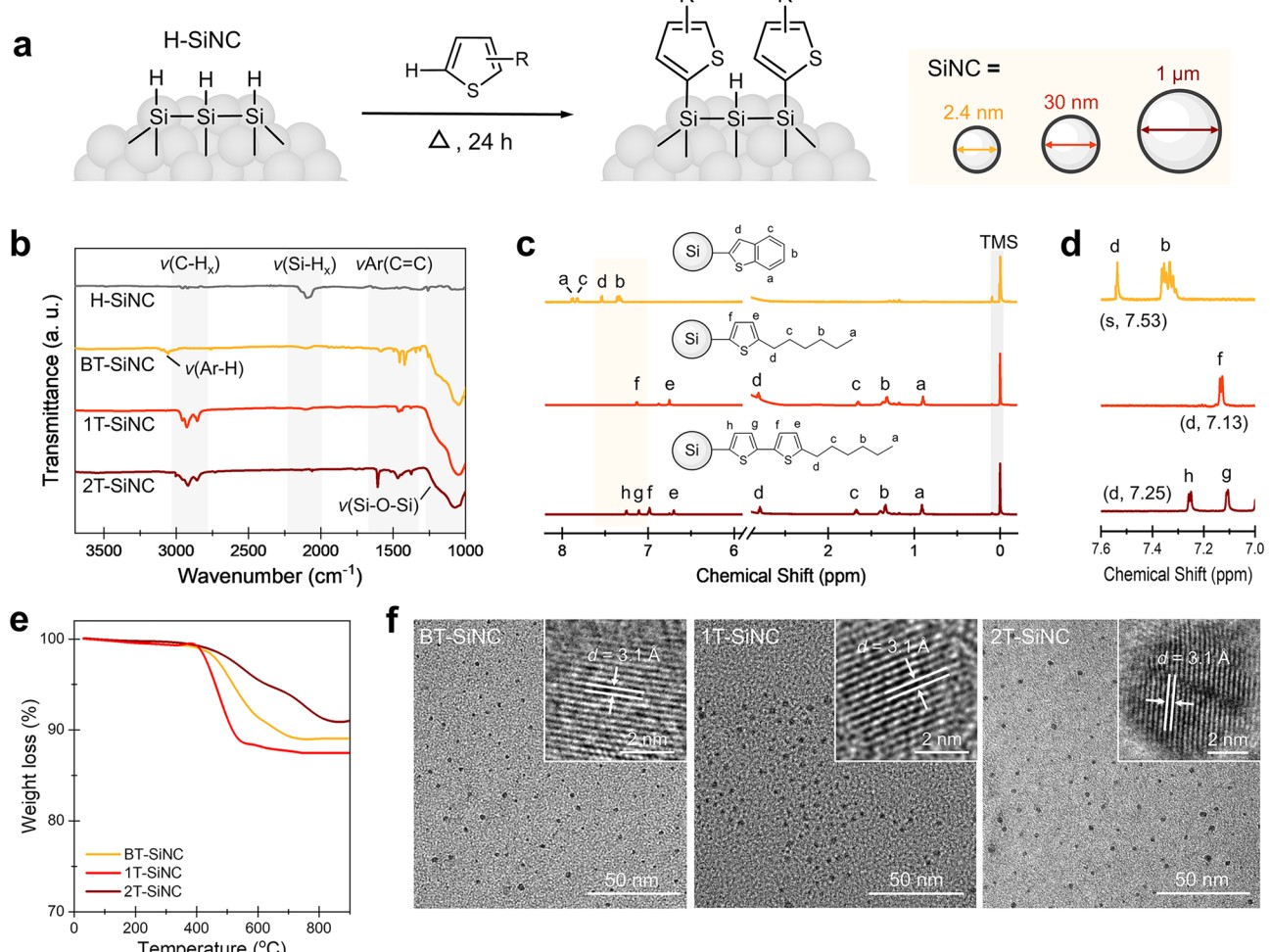

**Fig. 3 | Thermally driven dehydrocoupling reactions between thiophenes and silicon surfaces. a** Schematic illustration of the anchoring of thiophenes on the surfaces of silicon nanocrystals (SiNCs) with different sizes. **b** FT-IR and **c** solution-phase $^1$H NMR spectra of SiNCs ($d = 2.4$ nm) capped by BT, 1T, and 2T ligands (solvent: MeOD). Detailed analyses of the $^1$H NMR spectra are available in Supplementary Figs. 10–12. **d** Zoom-in NMR results highlighting the chemical shifting range between 7.0 and 7.6 ppm. **e** Thermal gravimetric analyzer (TGA) results of SiNCs ($d = 2.4$ nm) capped by BT, 1T, and 2T ligands. **f** Bright-field electron microscopy images of SiNCs functionalized with thiophenes shown in this figure (**a**).

calculation. The corresponding KIE values were then determined to be 2.73 and 2.67 for the reactions after 2 h and 4 h, respectively, indicating that the homolytic cleavage of the Si-H bond is the rate-determining step (RDS) throughout the reaction.

The characterizations presented above provide a comprehensive understanding of the dehydrocoupling reaction between thiophenes and molecular silanes (Fig. 2f). The reaction begins with the thermally induced homolytic cleavage of the Si-H bond, generating Si· and hydrogen radicals (H·). The Si· radicals then perform nucleophilic attack on the C2 position of the thiophenes through hydrosilylation process. Simultaneously, the radical site transfers from Si· to the C3 position of the thiophene ring. The H atom at the C2 position further interacts with the surrounding H·, leading to the elimination of gaseous hydrogen (see Supplementary Note 3 and Supplementary Fig. 9 for more details), re-aromatization of the thiophene ring, and the formation of Si-C linkage between the thiophenes and the molecular silane.

### Dehydrocoupling on silicon surfaces
This thermally driven dehydrocoupling reaction eliminates the need for high-boiling-point solvents or metal-based catalyst, which substantially simplify solvent/catalyst selection (some of which can oxidize Si-H bonds) and post-synthetic purification processes. Moreover,

the direct anchoring of highly conjugated thiophene species introduces new opportunities to tailor the optical and electronic properties of silicon materials. Motivated by the effectiveness of the reaction and these advantages, we sought to investigate whether this dehydrocoupling reaction could facilitate the direct anchoring of thiophenes on silicon surfaces. To investigate potential influence of the crystal size on reactivity, we selected colloidal silicon nanocrystals of three sizes (diameter ($d$) = 2.4 nm, ~30 nm, and ~1 μm) as model materials. All SiNCs were freshly etched with HF solution to yield hydride-terminated surfaces, rendering the particles dispersible in hydrophobic thiophenes, which served as both solvents and ligands (see "Methods" for experimental details). The mixture was heated at 180 °C under nitrogen to proceed with the dehydrocoupling reaction (Fig. 3a). The functionalized SiNCs were then isolated and purified for subsequent surface characterizations (see "Methods" for further details).

### Spectroscopic analysis of thiophene-anchored silicon nanocrystals
We firstly examined the effectiveness of the dehydrocoupling reaction on SiNC surfaces using BT as the ligand model. Regardless of the crystal size, all purified functionalized SiNCs have intense proton signals between 7 and 8 ppm in the solid-state $^1$H NMR spectra

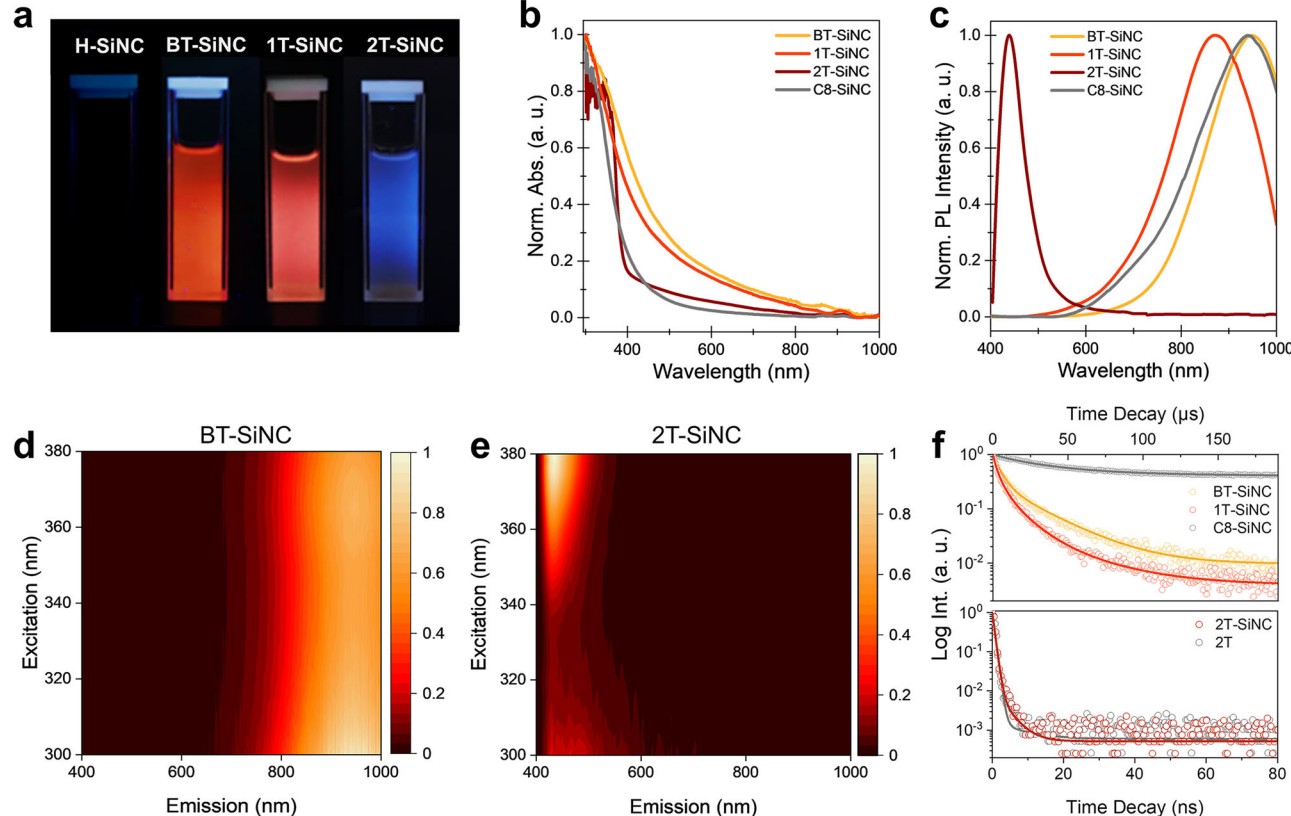

**Fig. 4 | Optical properties of thiophene-functionalized SiNCs. a** Photos of H-SiNCs and thiophene-functionalized products BT-SiNC, 1T-SiNC, and 2BT-SiNC dispersed in toluene under 365 nm UV illumination and their **b** absorption spectra and **c** photoluminescence spectra ($\lambda_{ex}$ = 360 nm). **d**, **e** Excitation–emission matrix (EEM) spectra of BT-SiNC and 2T-SiNC and **f** the comparison of their PL lifetime decay patterns with the control samples C8-SiNCs and 2 T molecule.

(Supplementary Fig. 13), indicative to the successful anchoring of BT on the silicon surfaces. Additionally, as the particle size increases, the spectral lines of the ligand signals broadened, and the resolution decreased. This can be attributed to changes in the local magnetic field (or dipole) interactions between nuclei[41,42].

To further confirm the generality of the surface dehydrocoupling reaction, we next applied 2.4 nm SiNCs to interact with 1T and 2T ligands and examine the surface modification using Fourier transform infrared spectroscopy (FT-IR). The spectrum of the original H-SiNCs has two distinctive prominent signals at ~2100 cm$^{-1}$ and ~850 cm$^{-1}$ corresponding to Si-H$_x$ ($x$ = 1–3) stretching and bending modes, respectively[43]. After interaction with thiophenes, the intensities of these signals diminished, and new features emerged in the ranges of 3200–3000 cm$^{-1}$, 2850–3000 cm$^{-1}$ and 1600–1450 cm$^{-1}$, which can be assigned to Ar-H$_x$, C-H$_x$, and aromatic skeletal stretching and bending modes of the anchored thiophenes (Fig. 3b). The conjugated ligand backbone and aliphatic side chains were further verified by solution-phase $^1$H NMR (Fig. 3c). An evident shift of the aromatic ring region (7–8 ppm) of the functionalized SiNCs compared to the freestanding ligand molecules was observed, which can be attributed to the shielding effect of the semiconducting SiNC core (Supplementary Figs. 10–12)[41,42]. The surface ligand coverage, estimated using thermogravimetric analysis (TGA), was similar across samples, ranging between 11 and 19% (Fig. 3e, Supplementary Note 4 and Supplementary Table 3). This is consistent with the presence of residual hydrides (Si-H$_x$) and suboxides (Si-O-Si) on the surfaces of the functionalized SiNCs (Fig. 3b). We attribute the incomplete surface passivation to the steric hindrance between the large aryl groups[14,44].

We next applied X-ray photoelectron spectroscopy (XPS) to investigate the elemental oxidation states of SiNCs following surface functionalization. All spectra confirmed that the functionalized SiNCs contained only silicon, carbon, and sulfur (Supplementary Figs. 15–17). High-resolution spectra revealed characteristic emissions at 99.3–99.4 eV, corresponding to the Si(0) signal from Si atoms in the SiNC cores[45]. Other features at 100.3–103.4 eV can be assigned to thiophene-linked surface Si atoms or suboxides[46]. Transmission electron microscopy (TEM) analyses confirmed that the functionalized SiNCs retained crystalline cores and share a similar size distribution of 2.4 ± 0.3 nm (Supplementary Fig. 14), indicating that the dehydrogenative functionalization process did not change the structure or size of the SiNCs.

**Optical and electronic properties of thiophene-capped particles**

Compared to phenyl rings, the high polarizability of thiophenes and their delocalized π electrons result in significantly higher charge carrier mobilities, potentially enabling the accumulation of charge carriers generated from the semiconducting core and tailoring the photophysical behaviors of SiNCs. Indeed, the thiophene-functionalized SiNC solutions exhibited remarkable differences in their photoluminescence (PL) under 365 nm irradiation (Fig. 4a): while the BT- and 1T-capped particles have red PL, 2T-capped SiNCs have intense blue emission. These observations suggest that radiative recombination occurs through different channels within these particles.

We therefore sought to systematically investigate the optical response of the functionalized SiNCs. Compared to the ligand molecules (Supplementary Fig. 18), all functionalized SiNCs show stronger absorption in the visible region (Fig. 4b), which are commonly observed from SiNCs functionalized with aliphatic or aromatic groups and attributed to the silicon-core-related absorption[47]. Prior to dehydrocoupling reactions, hydride-terminated SiNCs (H-SiNCs) exhibited no PL, likely due to the presence of abundant non-radiative surface

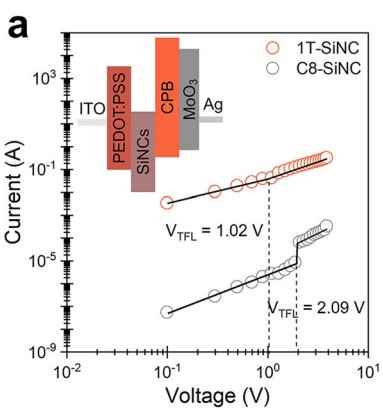
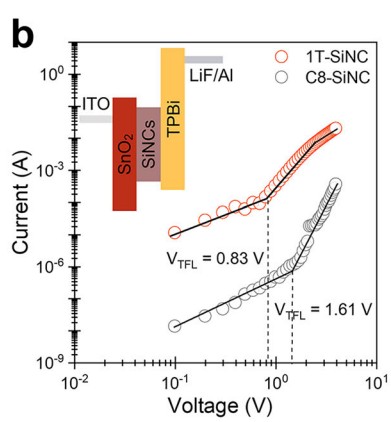
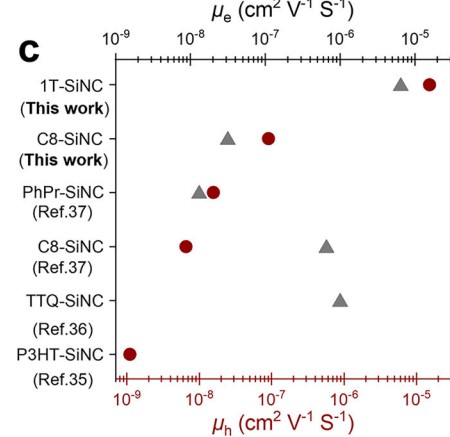

**Fig. 5 | Electrical properties of thiophene-functionalized SiNCs. a** Hole-only and **b** electron-only devices under dark conditions. Inset figures show the device architecture for both types of devices. **c** Comparison of carrier mobility results reported for functionalized SiNCs (Fig. 5c and Supplementary Table 10), high-lighting the potential of this dehydrocoupling approach to improve the optoelectronic properties of silicon-based devices. functionalized silicon nanocrystals. P3HT Poly(3-hexylthiophene-2,5-diyl), C8 Octyl, TTQ 2,3,5,6-Tetrafluoro-7,7,8,8-tetracyano-quinodimethane, PhPr Phenylpropyl.

traps or particle aggregation (Fig. 4a)[48,49]. After thiophene anchoring, SiNCs passivated with BT and 1 T demonstrated intense emission at 948 nm and 871 nm, respectively, with relatively broad full width at half-maximum (FWHM) of 220 nm (Fig. 4c). The corresponding excitation-emission matrix (EEM) spectra differed significantly from those of the ligand molecules (Fig. 4d, e and Supplementary Fig. 19), and microsecond-scaled PL lifetime decay were observed (BT-SiNCS: 206 µs; 1T-SiNCS: 156 µs, Fig. 4f).

For the 2T-capped SiNCs, a portion of the visible light absorption originated from the 2 T ligand, which primarily contributed by the transitions between the frontier molecular orbitals (Supplementary Note 5, Supplementary Figs. 20–23, and Supplementary Tables 6–8). After 2T anchoring, a new PL signal with a maximum at 438 nm emerged from the purified particles, with excitation and emission features highly similar to those of the freestanding 2T molecule (Fig. 4c and Supplementary Figs. 18 and 19). A nanosecond-scaled PL lifetime decay was also observed from 2T-SiNCs with an average lifetime of 0.8 ns, comparable to the that of the free ligand (0.7 ns, Fig. 4f and Supplementary Table 4). The emergence of the ligand-based blue emission of 2T-SiNCs suggests that the intrinsic band-edge radiative recombination was suppressed by 2T anchoring, which accumulates excitons from both the particles and ligands, serving as the radiative center for efficient blue emission (Supplementary Fig. 24). The photoluminescence quantum yield (PLQY) of 2T-SiNCs (6.4%) was significantly higher than that of the freestanding 2T molecule (3.2%) under the same measurement conditions ($\lambda_{ex}$ = 365 nm, Supplementary Table 5), indicating enhanced radiative recombination rates benefited by the efficient transfer of the charge carrier generated from the SiNC core[50,51].

In order to evaluate the impact of thiophenes on the electrical properties of SiNCs, we estimated charge carrier mobilities and trap densities using space-charge-limited current (SCLC) analysis in electron- and hole-only devices using thiophene-capped SiNCs as the active materials (details on device fabrication and calculations are available in "Methods"). The 1T-SiNC sample exhibited substantially improved solution processability compared to the other thiophene-capped samples. The corresponding exhibited lower trap density values of $8.8 \times 10^{15}$ cm$^{-3}$ ($n_{te}$) and $3.1 \times 10^{17}$ cm$^{-3}$ ($n_{th}$), compared to C8-capped SiNC counterpart ($1.7 \times 10^{16}$ cm$^{-3}$ ($n_{te}$) and $6.3 \times 10^{17}$ cm$^{-3}$ ($n_{th}$), Fig. 5a, b and Supplementary Table 9). This resulted in the enhancement of the charge carrier mobilities up to $6.5 \times 10^{-6}$ cm$^2$V$^{-1}$S$^{-1}$ ($\mu_e$) and $1.5 \times 10^{-5}$ cm$^2$V$^{-1}$S$^{-1}$ ($\mu_h$). These values are two to three orders of magnitude higher than those of C8-SiNC-based device (Supplementary Table 9) and surpass the performance of previously reported

In summary, we have developed a catalyst-free wet-chemistry approach for the direct linkage of thiophenes to organosilanes and silicon surfaces. This work leverages thermally driven homolytic cleavage of Si-H bonds to generate silyl radicals, which promote dehydrocoupling reactions to form Si-C with thiophene rings. The robust Si-C surface bonds and highly conjugated thiophene backbones enable enhanced charger carrier transport between crystalline silicon cores and surface ligands, yielding functionalized particles with tunable photophysical properties and reduced trap densities. This work demonstrates that catalyst-free dehydrocoupling reaction can be extended from molecular silanes to silicon surface functionalization, paving the way for efficient, highly processable, and cost-effective silicon-based optoelectronics.

## Methods

### Materials

All chemicals used are commercially available and were used without any additional purification steps: Trimethoxysilane (TMOS, 95%) was were purchased from Sigma-Aldrich Inc. Toluene (99%) was purchased from Guangzhou Chemical Reagent Inc. Hydrofluoric acid (HF, 49.99% aqueous solution, electronic grade) was purchased from Macklin Inc. Azobisisobutyronitrile (AIBN, 98%), n-hexane (≥98%), triphenylsilane (TPS, 98%), potassium t-butoxide ($^t$BuOK, 98%), 1-benzothiophene (BT, 98%), 2-hexylthiophene (1T, 98%), 5-hexyl-2,2′-bithiophene (2T, 97%), mesitylene (98%, Anhydrous, water ≤50 ppm), silicon powder (99.9%, 1 µm) were purchased from Aladdin Inc. Silicon powder (99.9%, 30 nm was purchased from Innochem. CDCl$_3$ (D, 99.8%, 0.03% v/v TMS), CD$_3$OD (D, 99.8%, 0.03% v/v TMS) were purchased from Cambridge Isotope Laboratories, Inc.

### Synthesis and purification of model molecules

**Benzo[*b*]thiophen-2-yltriphenylsilane (BT-TPS).** 147 mg of BTP (1.1 mmol) and 259 mg of TPS (1.0 mmol) were mixed and heated for 12 h in a pressure tube in the air. After the reaction, the ethyl acetate (EA) was added, giving the mixture. The mixture was passed through flash column chromatography (n-hexane is the eluent), yielding product BT-TPS (286 mg, 73% yield) as the white solid.

**(5-Hexylthiophen-2-yl)triphenylsilane (1T-TPS).** 184 mg of 1 T (1.1 mmol) and 259 mg of TPS (1.0 mmol) were mixed and heated for 12 h in a pressure tube in the air. After the reaction, the EA was added, giving the mixture. The mixture was passed through flash column

chromatography (n-hexane is the eluent), yielding product 1T-TPS (294 mg, 69% yield) as the white solid.

**Benzo[*b*]thiophen-2-yltriphenylsilane (2T-TPS).** 275 mg of 2T (1.1 mmol) and 259 mg of TPS (1.0 mmol) were mixed and heated for 12 h in a pressure tube in the air. After the reaction, the EA was added, giving the mixture. The mixture was passed through flash column chromatography (n-hexane is the eluent), yielding product 2T-TPS (320 mg, 63% yield) as the white solid.

### Synthesis of HSiO$_{1.5}$ sol-gel precursor

A 100-mL Schlenk flask was connected to a nitrogen-purged Schlenk line and fitted with a magnetic stirrer to facilitate the transfer of the necessary gases and ensure a thorough stirring action. 10 mL of Methanol (25 mmol) and 10 mL of 1% nitric acid (0.9 mmol) were added gradually and with stirring to the flask. Subsequently, 7.7 g of TMOS (19 mmol) was added under stirring conditions to initiate the hydrolysis. The reaction was conducted under nitrogen at room temperature. A colorless, transparent gel-like product had formed within three minutes, and stirring was immediately halted. The product was then left undisturbed for 24 h under a nitrogen atmosphere. After this aging period, the product was extracted from the residual liquid employing vacuum filtration and subsequently transferred to a vacuum oven, where it was subjected to drying for 16 h. During this process, the original colorless gel gradually transformed into a white, powdery substance, HSiO$_{1.5}$. The final product was then stored in a 20-mL glass vial at ambient conditions for subsequent utilization.

### Solid-state synthesis and liberation of SiNCs

A well-developed recipe was adopted to prepare hydride-terminated SiNCs[34]. Briefly, 1 g of the HSiO$_{1.5}$ powders were weighed out, placed in a quartz boat, and transferred to a tube furnace (Lindberg, TF55035KC-1). The sample was heated from room temperature to 1100 °C at 18 °C min$^{-1}$ in a slightly reducing atmosphere (5% H$_2$ + 95% Ar). The sample was held at the specified processing temperature for 1 h and subsequently cooled naturally to room temperature. The resulting SiNCs/SiO$_2$ product, which had a black-brown powdery composition, was then ground to a fine powder using an agate mortar. It was subsequently stored in a 20-mL vial and left in an atmosphere comprising air in order to facilitate its further utilisation.

### SiNCs or silicon powder were liberated using HF etching

0.2 g of the SiNCs/SiO$_2$ product or silicon powder was transferred to a 100-mL PVC beaker. A volume of 3 ml of water and 3 ml of ethanol were added to the beaker, with mechanical stirring maintained at room temperature throughout the addition. A 3-ml solution of HF in aqueous solution was then transferred to the beaker in order to initiate the reaction. **It should be noted that HF in aqueous solution must be handled carefully.** The reaction was stirred in ambient light at room temperature for one hour. After that, 3 × 20 mL of toluene was added to the beaker to extract the hydride-terminated SiNCs (H-SiNCs), and the upper solution was transferred via *a* - 5-mL plastic pipette. After the extraction, ~100 mL of saturated CaCl$_2$ aqueous solution was added to the PVC beaker to neutralize the residual HF. The extracted solution was centrifuged at 7830 rpm for 10 min. After that, the solution was decanted and the precipitate was transferred to a nitrogen-filled glove box for the next step.

### The reactions on hydride-terminated silicon surfaces with thiophene species

The abovementioned solution containing H-SiNCs or hydride-terminated silicon powders was diluted to 10 mL with 1,3,5-trimethylbenzene and transferred to a 15-mL thick-walled pressure tube and 0.5 g of predesigned thiophene compound was weighed from a nitrogen-filled glove box and transferred to the above pressure tube

and sealed. The sealed pressure tube was removed from the nitrogen-filled glove box and heated in an oil bath at 180 °C for 24 h. The solution gradually changed from turbid to clear. After the reaction tube was naturally cooled down to room temperature. The solution was transferred to 15-mL centrifuge tubes for centrifugation at 7830 rpm for 20 min. The supernatant was decanted, and the precipitate was redispersed with 10 mL of methanol. The centrifugation/precipitation/redispersion steps were repeated to remove the trace amount of unreacted ligands. After three times, the supernatant was decanted, and the precipitate was placed in a vacuum oven at 40 °C overnight to remove the residual 1,3,5-trimethylbenzene. That is, the corresponding surface-covered thiophene of SiNCs or silicon powder is obtained.

### Fourier transform infrared spectroscopy (FT-IR)

The thiophene-functionalized SiNC solution was centrifuged at 7830 rpm for 20 min. The supernatant was decanted, and the precipitate was transferred to a 15-mL vial and dried in a vacuum oven at 40 °C overnight. Afterward, the thiophene-functionalized SiNC powders were transferred onto the diamond sample window for the FT-IR measurements. The results were obtained using a Nicolet/Nexus-670 FT-IR spectrometer (ATR mode).

### XPS measurements

XPS measurements were carried out using a Thermo Scientific KAlpha XPS system with a monochromatic Al *K*α X-ray source (1486.7 eV, spot size: 400 μm). The electron kinetic energy was measured by an energy analyzer operated in the constant analyzer energy mode at 100 eV for elemental spectra. All spectra were internally calibrated to the C 1s emission (284.8 eV). Casa XPS software (ver. CasaXPS V2.3.16.PR1.6, VAMAS) was used to perform data fitting processes.

### $^1$H NMR and $^{13}$C NMR measurements

All NMR measurements were carried out using two Bruker Advance III NMR spectrometers (400 MHz and 600 MHz). Molecular silane products were tested with deuterated chloroform as solvent. ~5 mg of SiNC samples were transferred to an NMR tube using CD$_3$OD as the solvent. The NMR tube was ultrasonicated for ~1 min to form homogenous dispersion. Mestre Nova software (ver. 14.2.1, Mestrelab Research) was used to analyze the NMR results. All chemical shifts were recorded in parts per million (ppm, $\delta$) relative to residue chloroform (for $^1$H NMR, $\delta = 7.26$ ppm, singlet; for $^{13}$C NMR, $\delta = 77.16$ ppm, triplet) or residue methanol (for $^1$H NMR, $\delta = 3.31$ ppm, singlet). $^1$H NMR splitting patterns are designated as singlet (s), doublet (d), triplet (t), quartet (q), dd (doublet of doublets), m (multiplets), etc.

### Transmission electron microscopic (TEM) characterizations

TEM imaging was conducted using an FEI Talos F200s electron microscope, operating at an accelerating voltage of 200 kV. The thiophene-functionalised SiNCs were redispersed in toluene, and the supernatant was removed after 30 min of ultrasound treatment. The resulting solution was then drop-cast onto a 200-mesh carbon-coated copper TEM grid. Prior to the measurements, the TEM grid was subjected to a 16 h drying process in a vacuum oven at 25 °C to eliminate residual solvent. All images were processed utilising the ImageJ software (ver. 1.52a).

### Measurements of static-state photoluminescent properties of SiNCs

Excitation-emission matrix (EEM) and absorption spectra of the SiNC toluene solutions were obtained using a Duetta spectrometer (Horiba Inc.). The results of the PL lifetime decay were obtained using an FLS980 fluorescence spectrometer (Edinburgh Instruments), which was equipped with a Xenon lamp and a monochromator. The PL lifetime decay results were obtained using a 369.6 nm pulse laser as the excitation source, generated by a picosecond pulsed diode laser station (Edinburgh Instrument Ltd., EPL 369.6 nm).

### Single-crystal X-ray diffraction

The single-crystal X-ray diffraction data was recorded at 223 K on an Agilent Gemini Ultra diffractometer with CuKα radiation ($\lambda = 1.54184$ Å). The relative configuration of product BT-TPS was assigned based on the crystal X-ray. Vaporization of the $n$-hexyne solution of compound BT-TPS obtained a colorless needle crystal of BT-TPS. The relative structure of BT-TPS was solved by SHELXT methods and refined using full-matrix least-squares difference Fourier techniques.

### Analysis of instantaneous rate of reaction in kinetic process

The accumulated time-dependent yield generation data were subjected to a pseudo-first order kinetic equation to obtain a fitted result[52]:

$$\ln[q_e - q_t] = \ln q_e - k_1 t \qquad (1)$$

By rearranged to obtain,

$$q_t = q_e[1 - \exp(-k_1 t)] \qquad (2)$$

In this context, $k_1$ represents the rate constant of the reaction, while $q_e$ denotes the quantity of formation product at equilibrium and $q_t$ signifies the amount of yield at any given time point $t$. The time-dependent product generation rate outcomes were derived through the application of the first-order derivative to the corresponding time-dependent product generation curves.

### KIE study of thermally driven dehydrogenative coupling process

A well-developed recipe was adopted to prepare TPS(D)[53]. 147 mg of BT (1.1 mmol) and 259 mg of TPS (1.0 mmol) were mixed and heated in a pressure tube for 2 h and 4 h in the air. Similarly, 147 mg of BT (1.1 mmol) and 261 mg of TPS(D) (D, 99%, 1.0 mmol) were mixed and heated under identical conditions. After the reaction, 2 mL of EA was added to the mixture, which was then purified using flash column chromatography with n-hexane as the eluent.

### Fabrication of the hole- and electron-only devices

The ITO-coated glass substrates (1.48 cm × 1.48 cm) were cleaned consecutively with Triton X-100 solution, deionized water, ethanol, acetone, and isopropanol using sonication for 30 min each. The substrates were dried under continuous $N_2$ flow and treated with plasma at 80 W for 15 min prior to the device fabrication.

For electron-only devices, ~80 μL of SnO$_2$ NC solution (3–4 wt.% dispersion in H$_2$O) was initially spin-coated on the clean ITO substrate at 3000 rpm for 40 s and subsequently annealed at 150 °C for 15 min under atmosphere conditions. After that, the substrates were naturally cooling to room temperature. 45 μL of ~30 mg mL$^{-1}$ SiNC toluene solution was spin-coated on the substrate at 2000 rpm for 40 s in the nitrogen-filled glovebox, followed by thermal annealing at 150 °C for 30 min. The resulting thin films were then naturally cooled to room temperature, and transferred to a nitrogen-filled glovebox. As for top electrode, 40 nm of TPBi, 1 nm of LiF and 100 nm of Al were thermally deposited on the SiNC layer to complete the device (vacuum pressure $<5 \times 10^{-4}$ Pa).

For hole-only devices, 100 μL of PEDOT:PSS aqueous solution (1.3–1.7 wt.% dispersion in H$_2$O) was spin-coated on a clean ITO substrate at 4000 rpm for 40 s under ambient conditions, followed by thermal annealing at 150 °C for 30 min and the subsequent natural cooling to room temperature. Subsequently, 45 mL of 30 mg mL$^{-1}$ SiNC toluene solution was spin-coated on top of the PEDOT:PSS layer at 2000 rpm for 40 s, followed by thermal annealing at 150 °C for 30 min. The resulting thin films were then cooled to room temperature, and transferred to a nitrogen-filled glovebox. For device completion, 60 nm of CPB, 6 nm of MoO$_3$, and 100 nm of Ag were thermally deposited on the SiNC layer to complete the device (vacuum pressure $<5 \times 10^{-4}$ Pa). Each ITO substrate was patterned to yield four devices, each with an active area of 8.0 mm$^2$.

### Determination of carrier mobility and trap density values using space-charge-limited current (SCLC) measurements

The characterizations were carried out at room temperature in a nitrogen-filled glovebox. Current density-voltage (J-V) characteristics were recorded by Keithley 2400 source meter with a step of 0.1 V (19 ms per step).

The carrier mobility values were calculated using Mott–Gurney's equation:

$$\mu = \frac{8 J_D d^3}{9 \varepsilon \varepsilon_0 V^2} \qquad (3)$$

where $J_D$ is the current density, $d$ is the thickness of the SiNC film (determined by the cross-sectional SEM images shown in Supplementary Figs. 25 and 26), $\varepsilon_O$ is the vacuum dielectric constant, and $\varepsilon = 9.8$ is the relative dielectric constant for SiNC[54].

And the trap densities of electron and hole can be calculated using the following equation:

$$n_{t(e/n)} = \frac{2 \varepsilon \varepsilon_0 V_{TFL(\frac{e}{n})}}{ed^2} \qquad (4)$$

where $n_{t(e/h)}$ is the trap state density of electron/hole, $V_{TFL}$ is the trap-filled limit voltage, and $e$ is the elementary charge.

## Data availability

All the data supporting the findings of this study are available within the article and its Supplementary Information or from the corresponding authors upon request. The X-ray crystallographic coordinate for the structure reported (BT-TPS) data generated in this study have been deposited in the Cambridge Crystallographic Data Centre (CCDC) database under accession code CCDC 2361418 [https://www.ccdc.cam.ac.uk/].

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

## Acknowledgements

The authors acknowledge funding from the National Natural Science Foundation of China (22175201 (Z.Y.), 22475239 (Z.Y.)), the Pearl River Recruitment Program of Talent (2019QN01C108 (Z.Y.)), the Guangzhou Science and Technology Programme (2024A04J6360 (Z.Y.)), the Guangdong Basic Research Center of Excellence for Functional Molecular Engineering (2024G0003 (Z.Y.)), and Sun Yat-sen University. We thank Prof. Jing Wang and Dr. Lin Huang for assistance with PL measurements.

## Author contributions

Z.Y. conceived the idea and directed this study. J.L. led the experimental work and developed the synthetic recipe, prepared the materials, and

carried out the characterizations and analysis. M.Z. fabricated the devices, performed the tests, and analysed the data. W.L. carried out the single-crystal characterizations and analysed the data. Z.L. performed the computational study. J.L., T.Z., and Z.Y. analysed the NMR results, drafted and finalized the reaction mechanism. J.L. and Z.Y. wrote the manuscript. All authors read and commented on the manuscript.

## Competing interests

A provisional patent application CN2024109891132 was filed on July 23, 2024 by the Sun Yat-sen University. The remaining authors declare no competing interests.
