## [Transparent Peer Review file · Nature Communications]

Direct Dehydrocoupling Facilitates Efficient Thiophene Anchoring on Silicon Surfaces

Corresponding Author: Professor Zhenyu Yang

Version 0:

Reviewer comments:

Reviewer #1

(Remarks to the Author)

The manuscript titled "Direct Dehydrocoupling Facilitates Efficient Thiophene Anchoring on Silicon Surfaces" describes a novel approach in silicon surface chemistry by enabling the direct anchoring of thiophene onto silicon surfaces through catalyst-free dehydrocoupling. The proposed approach differs from existing methods that often require pre-activation of precursors or transition-metal catalysts. The reported enhancement in charge carrier mobility in silicon nanocrystals by 10–100 times compared to conventional alkyl-functionalized counterparts emphasized the potential impact of the current method in advancing silicon-based optoelectronic materials. Overall, this is a well-designed and significant study. Given the novelty of the approach and the significant improvement in material properties, this work appears to be original and a valuable contribution to the field, and therefore I recommend its publication after minor revision. Below are my comments and suggestions to enrich the content:

The authors have previously investigated dehydrocoupling reactions on silicon surfaces, as reported in their recent publication "Nucleophilic Attack Enables Crystalline Silicon Formation Through Dehydrocoupling at Room Temperature." While the current manuscript explores a different bonding strategy (Si–C bond formation via thiophene anchoring), the earlier work represents an important piece of work in this area. I recommend citing this earlier study to better understand the evolution of the authors' research. It would improve the story of methodological progress (shift from Si–Si to Si–C bonding) and make it more clearly by highlighting the role of this study in the larger field of silicon surface functionalization.

The authors provide extensive spectroscopic and microscopic evidence supporting the successful thiophene functionalization of SiNCs. Could the authors elaborate on how they distinguished the covalent Si–C bonding from possible physisorption surface interactions?

On Page 15, the authors state that: Prior to dehydrocoupling reactions, hydride-terminated SiNCs (H-SiNCs) exhibited no PL, likely due to the presence of abundant non-radiative surface traps or particle aggregation (Figure 4a). Could the authors elaborate further on the absence of PL by clarifying why the H-SiNCs appear to lack sufficient surface passivation and why the quantum confinement effect does not seem to be effectively realized in this case?

The observed ligand-dominated PL in 2T-functionalized SiNCs, along with blue-shifted emission and short PL lifetime, suggests the possible involvement of Förster Resonance Energy Transfer (FRET) from the SiNC core to the 2T ligand. I recommend authors to address this and briefly discussing the possibility.

Fig.3(c), solution-phase ^1H NMR spectra, Unknown impurities were denoted using triangles (\blacktriangle). Can the authors speculate on the nature of these impurities (oxidation, side products, ...)?

On Page 6, After 12 h, the reaction yielded BT-TPS at 73%. Could the authors comment on whether prolonged reaction times beyond 12 h affect product yield or purity?

In Figure 2c and Figure 4b, 4c: Please consider changing/revising the color choices to better improve contrast and readability of the graphs.

Grammatical errors: "the modification the surfaces" should be "the modification of the surfaces".

"This is typically achieved replacing chemically active bonds..." should be "This is typically achieved by replacing chemically active bonds..."

"hydroxtlation" should be "hydroxylation"

Please rephrase the sentence in (Page 17) to avoid repetition (efficient charger carriers transport) and (improved charger carrier mobilities).

Farid A. Harraz, PhD
Professor

Reviewer #2

(Remarks to the Author)

This paper reports the development of the direct covalent linkage of thiophenes to silicon surfaces, developed from a model using a solution based silanes mimic. The m/s begins with a comprehensive study and optimisation of the solution reaction of Ph₃SiH and benzothiophene. Although the reaction proceeds best at 180°C, given the nature of the amorphous Si substrate to be used later, this is not an issue. The presence of radical intermediates was demonstrated using radical inhibiting agents, and kinetic isotope studies confirmed the location of and RDS for the reaction. The authors make the case that this dehydrocoupling reaction eliminates the need for high-boiling solvents and catalyst, and is ideally suited for solid state application. The solution process was then systematically applied to Si substrates, and surface modification demonstrated by a combination of NMR, XPS, and luminescence measurements. The optical response and electrical properties of modified particles were measured and the modified particles shown to possess reduced trap densities and improved charge carrier mobilities.

The methodology used in the work is appropriate and has been clearly delineated in the m/s, and the authors have elaborated a clever effective method for the surface modification of Si substrates, in a clearly written paper which has been produced to an exceptionally high standard. This work is likely to be of considerable importance and interest to material and surface scientists, and is worthy of publication.

The m/s is nearly error free (a couple of very minor typos are indicated on the attached file), and I suggest that the authors use the XPS data to calculate C-H ratios, and compare these to the theoretical value for thiophene, as further evidence of the claimed surface modification.

With these corrections made, I consider that the m/s is suitable for immediate publication.

Reviewer #3

(Remarks to the Author)

This manuscript presents a catalyst-free radical strategy that leverages thermally induced homolytic cleavage of Si–H bonds to generate reactive silicon radicals, which subsequently undergo direct hydrosilylation with thiophene rings to form stable Si–C linkages. The authors further investigate the interaction of various ligands with silicon nanocrystals, not only to demonstrate the generality of the approach but also to highlight its influence on the optical properties of the resulting materials. The synthesis procedures are described in detail, and the experimental data are comprehensive and support the authors' conclusions.

Overall, this work is compelling and, in my opinion, well-suited for publication in Nature Communications after minor revision. It introduces a mild, metal-free method for silicon surface functionalization, applicable to both silicon nanocrystals and bulk substrates such as wafers. This versatile strategy meaningfully expands the synthetic toolbox for silicon-based hybrid materials and holds strong potential for applications in optoelectronics.

It would be great if the authors could clarify the following points or provide additional discussion .

1. Regarding the blue emission of 2T–Si NCs, it appears to closely resemble the emission of the thiophene molecule itself. Could this be attributed to strong electronic coupling between the molecule and the nanocrystal, leading to rapid energy transfer from the ligand to the Si NC? A more detailed discussion on this possible mechanism would be valuable.
2. In Fig. 4h, there is a sharp increase in the current near the VTFL (trap-filled limit) point. Could the authors provide further insights or hypotheses to explain this behavior?
3. In Fig. 4 and SI Fig.S19, the excitation wavelength range stops at 380 nm. I am curious—what photoluminescence response would be expected from the 2T–Si NCs if the excitation were extended to longer wavelengths, such as 500 nm or 600 nm?
4. TEMPO was used during synthesis to support the radical-mediated mechanism and to probe the determining step. Have the authors considered using electron paramagnetic resonance (EPR) spectroscopy to directly detect the quenching of radicals by TEMPO, which would provide more direct evidence for the proposed mechanism?
5. As the authors mentioned that the ligand loading is not complete, it would be helpful to discuss possible strategies or future plans to improve ligand coverage, which could be important for further tuning of surface properties and device performance.
6. Regarding the ligand coverage, it would be helpful if the authors could provide a step-by-step explanation of the calculation method in SI.
7. In Fig. 3e, the nanocrystal diameter is reported as approximately 3 nm, whereas in the Supporting Information, the size is noted to be in the range of 2.3–2.5 nm. Please double check.

Version 1:

Reviewer comments:

Reviewer #1

(Remarks to the Author)

I have carefully checked the revised version of the manuscript. The authors have adequately addressed the comments and suggestions raised by reviewers. The overall quality of the content has improved significantly, and the manuscript is now suitable for publication in its current form.

Farid A. Harraz, PhD
Professor
Advanced Materials and Nano-Research Centre
Najran University, Saudi Arabia
faharraz@nu.edu.sa

Reviewer #2

(Remarks to the Author)

I am content with the authors response to my suggestion, and with the corrections which they have made

Reviewer #3

(Remarks to the Author)

The authors have addressed all of my questions. I recommend publishing without further revision.

Reviewer#1 (Remarks to the Author):

1. The authors have previously investigated dehydrocoupling reactions on silicon surfaces, as reported in their recent publication "Nucleophilic Attack Enables Crystalline Silicon Formation Through Dehydrocoupling at Room Temperature." While the current manuscript explores a different bonding strategy (Si–C bond formation via thiophene anchoring), the earlier work represents an important piece of work in this area. I recommend citing this earlier study to better understand the evolution of the authors' research. It would improve the story of methodological progress (shift from Si–Si to Si–C bonding) and make it more clearly by highlighting the role of this study in the larger field of silicon surface functionalization.

Response: We thank the reviewer for the recommendation for publication and the recognition of the novelty of our previous and current studies. Our team have been investigating various types of silicon-based reactions and the corresponding applications on synthesis and surface functionalization of silicon materials. We have followed the reviewer's suggestion, and cited the relevant on the construction of silicon-based materials based on different bonding strategy in the introduction. Now it reads:

*"...The versatility of silicon extends to surface chemistry, where its ability to form covalent bonds with various elements enables the modification of the surface silicon atoms on silicon-containing solids and even the construction of new silicon-based structures and hybrid materials. (REF: Adv. Funct. Mater. **24**, 1345–1353, (2014); CCS Chemistry DOI: 10.31635/ccschem.024.202405067, (2024)) This is typically achieved by replacing chemically active bonds..."*

2. The authors provide extensive spectroscopic and microscopic evidence supporting the successful thiophene functionalization of SiNCs. Could the authors elaborate on how they distinguished the covalent Si–C bonding from possible physisorption surface interactions?

Response: We thank the reviewer for this important question regarding the distinction between covalent Si-C bonding and physisorption. All spectroscopic techniques used to confirm ligand anchoring rely on the fundamental premise that the ligands are chemically linked to SiNC surfaces through Si-C bonds rather than through physical adsorption.

Unlike the ionic bonding between ligands (e.g., oleylamine, oleic acid) and the surfaces of ionic quantum dots (e.g., CdSe, PbS), which can be cleaved during purification, the covalent Si-C bonds on functionalized SiNCs are robust enough to withstand multiple washing steps without detaching from the surface (REF: *Chem. Rev.* **102**, 1273, (2002); *Angew. Chem. Int. Ed.* **55**, 2322-2339, (2016)). In this study, we implemented a multi-step purification process including three rounds of washing with hydrophobic solvents to effectively remove unbound ligand molecules. This approach is well-established in silicon surface chemistry research (REF: *J. Am. Chem. Soc.* **134**, 489–497, (2012); *J. Am. Chem. Soc.* **138**, 8639–8652, (2016); *J. Am. Chem. Soc.* **139**, 5870–5876, (2017); *Nature Chemistry* **12**, 137–144, (2020)). After these rigorous separation and purification steps, we detected no thiophene signals in the ¹H-NMR analysis of the mother liquor. Therefore, we conclude that the ligands remaining on the washed SiNCs are chemically bonded to the surfaces.

3. On Page 15, the authors state that: Prior to dehydrocoupling reactions, hydride-terminated SiNCs (H-SiNCs) exhibited no PL, likely due to the presence of abundant non-radiative surface traps or particle aggregation (Figure 4a). Could the authors elaborate further on the absence of PL by clarifying why the H-SiNCs appear to lack sufficient surface passivation and why the quantum confinement effect does not seem to be effectively realized in this case?

Response: The lack of observable PL on H-SiNCs arises from the intrinsic instability of surface Si-H bonds that commonly get oxidized and create non-radiative traps that quench the PL. The relatively weak Si-H bonds (~3.4 eV bond energy vs. 4.5 eV for Si-C bond) provide inadequate protection against surface oxidation and dangling bond formation, even in freshly prepared samples (REF: *Sci. China Chem.* **66**, 1654–1687, (2023)). These surface defects create deep trap states (~0.2 – 0.4 eV) that dominate carrier dynamics through rapid non-radiative recombination (REF: *Nano Lett.* **20**, 1952-1958, (2020); *J. Appl. Phys.* **120**, 145302, (2016)). While quantum confinement effects exist in the crystalline core of nanostructures, their electronic signatures are completely masked by surface potential fluctuations that broaden energy levels and facilitate non-radiative recombination (REF: *Phys. Chem. Chem. Phys.*, **24**, 13519-13526, (2022)). This explains why even monodispersed H-SiNCs typically show no detectable PL, unlike their thiophene-passivated counterparts where robust Si-C bonding eliminates these non-radiative pathways. In addition, the limited dispersity of H-SiNCs in hydrophobic solvents due to the lack of surface ligands that cause the inevitable particle aggregation in solution that quenches the PL, which is also found in our previous studies (REF: *ACS Nano* **16**, 15450–15459, (2022); *Angew. Chem. Int. Ed.* **62**, e202304056, (2023)).

4. The observed ligand-dominated PL in 2T-functionalized SiNCs, along with blue-shifted emission and short PL lifetime, suggests the possible involvement of Förster Resonance Energy Transfer (FRET) from the SiNC core to the 2T ligand. I recommend authors to address this and briefly discussing the possibility.

Response: We appreciate the reviewer's suggestion regarding FRET in the 2T-SiNC system. While the observed significant blue-shifted emission and notably shortened PL lifetime might initially suggest FRET process, spectral analyses reveal this mechanism is unlikely in our system:

FRET requires a significant overlap between the PL of the energy donor and the absorption of the energy acceptor (*Chem. Soc. Rev.*, **49**, 5110—5139, (2020)). Previous studies about FRET on SiNC systems indeed meet the requirement of the PL/absorption overlapping, in which the reported SiNCs emit in the blue region (REF: *J. Biomed. Opt.* **22**, 087002, (2017); *Nanoscale*, **4**, 5163, (2012); *Nanoscale*, **15**, 12492, (2023)). In our study, however, the near-infrared (NIR) band-edge emission of SiNCs (e.g., C8-SiNC's emission at ~900 nm) shows negligible overlap with the absorption spectrum of 2T ligands (Figure R1, see below). Because our system differs with the fundamental requirement for FRET, FRET is unlikely to occur on 2T-SiNCs.

Figure R1. Absorption and emission spectra of freestanding 2T molecule, and the emission spectrum of C8-SiNC.

Instead, we attribute the blue emission contributed by the anchored 2T ligands which serve as the new radiative center. This is not only evidenced by the similar EEM spectra between the 2T-passivated particles and the freestanding 2T molecules (Figure 4e and Supplementary Figure 19), but also evidenced by the enhanced PLQY of the blue emission on 2T-SiNCs compared to the freestanding molecule solution. These phenomena are consistent with the photophysical features of reported SiNCs directly linked with conjugated ligands (REF: *Nano Lett.* **15**, 3657-3663, (2015), *Phys. Chem. Chem. Phys.* **16**, 19275-19281, (2014), and *Angew. Chem. Int. Ed.* **62**, e202304056, (2023)).

5. Fig.3(c), solution-phase ^1H NMR spectra, Unknown impurities were denoted using triangles (\blacktriangle). Can the authors speculate on the nature of these impurities (oxidation, side products, ...)?

Response: After careful re-examination, we determined that the signals labelled with triangles in original Figure 3c likely arose from trace solvent residues or reagent cross-contamination rather than reaction byproducts (e.g., oxidation or side products). To ensure the highest data quality, we repeated the synthesis with optimized workup procedures and obtained new NMR spectra under identical conditions and the updated results are shown in the updated Figures 3c, 3d, and Supplementary Figures 10 – 12. Now these spectra present impurity-free spectra, which confirm the purity of the products.

6. On Page 6, After 12 h, the reaction yielded BT-TPS at 73%. Could the authors comment on whether prolonged reaction times beyond 12 h affect product yield or purity?

Response: We thank the reviewer for raising the question about reaction optimization. To address this, we conducted extended reactions (16 h vs. 12 h as the standard process shown in the original work) and observed comparable yields (~73%) with no significant difference in product purity, as confirmed by highly similar ^1H -NMR spectra (Figure R2, see below). This suggests the reaction reaches completion within 12 h, with no observable difference from prolonged heating (e.g., no

additional byproducts or improved conversion), likely due to the thermodynamic equilibrium being established by 12 h.

Figure R2. ^1H NMR spectra of BT-TPS after dehydrocoupling reaction for (a) 12 h and (b) 16 h. NMR parameters are similar to those shown in Supplementary Figure 2.

We would like to clarify that the 73% yield reflects the isolated product mass after flash column purification (accounting for typical processing losses), while the 95% conversion in Figure 2c is calculated from the disappearance of Si-H signals in ^1H NMR. Notably, both 12-hour and 16-hour reactions achieved $\sim 95\%$ conversion by NMR analysis, confirming comparable reaction completeness. We have added explicit methodological notes in the revised manuscript:

(Regarding Yield): "...yielding the desired aromatic silane product BT-TPS with a 73% yield (Figure 2a, measured by the weight of purified product after column chromatography) ..."

(Regarding Conversion): "...Figure 2c. Kinetic study of the dehydrocoupling reaction between BT and TPS at various temperatures. Conversion values were calculated based on Si-H signal attenuation in ^1H NMR spectra..."

7. In Figure 2c and Figure 4b, 4c: Please consider changing/revising the color choices to better improve contrast and readability of the graphs.

Response: We have followed the reviewer's suggestion and updated Figures 2c, 4b, and 4c with higher contrast colors for better readability.

8. Grammatical errors: “the modification the surfaces” should be “the modification of the surfaces”.

Response: We appreciate the reviewer catching this error. The text has been corrected to “*the modification of the surfaces...*” in the revised manuscript.

9. “This is typically achieved replacing chemically active bonds...” should be “This is typically achieved by replacing chemically active bonds...”

Response: We have corrected the sentence as suggested “*This is typically achieved by replacing chemically active bonds...*”.

10. “hydroxtlation” should be “hydroxylation”

Response: We appreciate the reviewer catching the error. The typo has been fixed in the revised manuscript.

11. Please rephrase the sentence in (Page 17) to avoid repetition (efficient charger carriers transport) and (improved charger carrier mobilities).

Response: We thank the reviewer for this helpful suggestion. We have revised the paragraph on Page 17 (i.e., Summary section) to eliminate the redundant phrases regarding charge transport properties, while maintaining all key scientific information. Now it reads:

“...The robust Si-C surface bonds and highly conjugated thiophene backbones enable enhanced charger carrier transport between crystalline silicon cores and surface ligands, yielding functionalized particles with tunable photophysical properties and reduced trap densities.”

Reviewer#2 (Remarks to the Author):

1. The m/s is nearly error free (a couple of very minor typos are indicated on the attached file), and I suggest that the authors use the XPS data to calculate C-H ratios, and compare these to the theoretical value for thiophene, as further evidence of the claimed surface modification. With these corrections made, I consider that the m/s is suitable for immediate publication.

Response: We appreciate the reviewer's positive assessment of our manuscript and the constructive suggestion regarding XPS analysis. We have carefully refitted the high-resolution C 1s spectra to identify different carbon environments in our functionalized SiNCs (Figure R3, summarized below). The observed binding energies align well with reported values for similar SiNC systems (REF: *Adv. Funct. Mater.* **31**, 2008708, (2021); *Adv. Energy Mater.* **8**, 1701580, (2018)).

While we agree that XPS can provide valuable qualitative information about carbon species, we have chosen not to perform quantitative C-H ratio analysis due to the inevitable presence of adventitious carbon contamination. This contamination, commonly observed at ~284.8 eV (C-H/C-C), can originate from multiple sources in our system including residual solvents and vacuum pump oils. Given these limitations, we believe NMR spectroscopy provides more reliable quantitative information about the ligand bonding structure, while the XPS data serves as excellent complementary evidence for surface modification.

Figure R3. High-resolution XPS results in the region of carbon 1s region: (a) BT-SiNC, (b) 1T-SiNC, and (c) 2T-SiNC. Related images have been also updated in Supplementary Figure 15 – 17).

Reviewer#3 (Remarks to the Author):

1. Regarding the blue emission of 2T–Si NCs, it appears to closely resemble the emission of the thiophene molecule itself. Could this be attributed to strong electronic coupling between the molecule and the nanocrystal, leading to rapid energy transfer from the ligand to the Si NC? A more detailed discussion on this possible mechanism would be valuable.

Response: This question is relevant to the Question#4 of Reviewer#1. We agree with the reviewer that the emission of 2T-SiNC seems originate from the anchored 2T ligands. To better address the reviewer's question, we systematically compared the absorption and PL features of 2T-SiNC and freestanding 2T molecule. The results shown in Figure 4b and updated Supplementary Figure 18 reveals significant spectral overlap, making it challenging to definitively establish energy transfer mechanisms under standard excitation conditions (i.e., excitation at 360 nm in the manuscript).

To specifically probe this question, we conducted additional experiments using a narrow-band 450 nm laser (EPL-450, Edinburgh Instruments), which preferentially excites the ligand moiety while minimizing direct SiNC excitation (Figure R4). Control measurements of toluene solvent and instrument background confirmed that the observed signals at 438 nm (PL of freestanding 2T ligand) were absent in 2T-SiNC samples under these conditions, and we detected only weak scattering signals from particle aggregates. These results suggest that the excitation efficiency at 450 nm may not be sufficient to populate emissive states, or surface trapping dominates the relaxation pathway under these excitation conditions. Future studies employing stronger excitation sources and improved particle dispersion (e.g., through multi-step functionalization as discussed in Response#5 to Reviewer #3) may help clarify these dynamics.

Figure R4. (a) PL spectra of 2T-SiNC toluene solution, clean toluene (solvent), and the instrumental blank under 450 nm laser excitation, (b) Laser operation parameters (wavelength and bandwidth) obtained from EPL-450 Laser Instruction (Edinburgh Instruments Inc.).

2. In Fig. 4h, there is a sharp increase in the current near the VTFL (trap-filled limit) point. Could the authors provide further insights or hypotheses to explain this behavior?

Response: The sharp current increase near V_{TFL} in Figure 4h represents a characteristic transition in space-charge-limited current (SCLC) transport. Below V_{TFL} , conduction occurs primarily through trap-assisted hopping, where injected carriers are progressively captured by deep-level traps, leading to space charge accumulation. When the applied voltage reaches V_{TFL} , all available trap states become completely filled, triggering an abrupt transition to trap-free conduction. The saturation of trap states eliminates localized pathways through the bandgap, causing carrier mobility to surge from trap-limited values to intrinsic levels as the system enters the Child's law regime. The resulting conductance increase of 3 – 4 orders of magnitude explain the observed sharp inflection in the I - V features shown in Figure 4h (REF: *ACS Energy Letters* **5**, 376–384, (2020); *ACS Energy Lett.* **6**, 1087–1094, (2021)).

3. In Fig. 4 and SI Fig.S19, the excitation wavelength range stops at 380 nm. I am curious—what photoluminescence response would be expected from the 2T–Si NCs if the excitation were extended to longer wavelengths, such as 500 nm or 600 nm?

Response: As noted in our response to Question#1 of Reviewer#3, we performed additional experiments using 450 nm laser excitation (EPL-450, Edinburgh Instruments) to probe this behavior. Unfortunately, we observed no characteristic blue PL emission from 2T-SiNCs under these conditions. This absence of signal likely results from either (1) insufficient absorption at longer wavelengths, or (2) rapid exciton trapping at surface states that prevents radiative recombination through the 2T ligands. While we could not test excitation at 500-600 nm due to equipment limitations, the 450 nm results suggest that extending the excitation range further into the visible spectrum would not yield the characteristic blue emission.

4. TEMPO was used during synthesis to support the radical-mediated mechanism and to probe the determining step. Have the authors considered using electron paramagnetic resonance (EPR) spectroscopy to directly detect the quenching of radicals by TEMPO, which would provide more direct evidence for the proposed mechanism?

Response: We accepted the reviewer's concern about the direct evidence about the function of TEMPO for quenching the radical-driven dehydrocoupling reaction. We returned to the laboratory, proceeded two parallel samples (with or without the addition of TEMPO) for the reactions between TPS and BT molecules, and conducted comparative in situ electron paramagnetic resonance (EPR) measurement at 453 K for both samples (Figure R5, see below). The control reaction without TEMPO exhibited a characteristic broad singlet signal at $g = 2.0059$, which we attribute to the formation of triphenylsilyl radicals under our reaction conditions, consistent with literature reports (REF: *Appl. Magn. Reson.* **18**, 425–434, (2000)). While in the presence of TEMPO, this silyl radical signal was completely absent, with only the distinctive TEMPO radical signal ($g = 2.0077$) observable (REF: *J. Chem. Phys.* **126**, 044512, (2007)). These EPR results provide direct experimental evidence supporting our proposed mechanism where TEMPO effectively scavenges

the reactive silyl radical intermediates, ultimately forming TEMPO-incorporated silanol structures as further confirmed by our HRMS analysis (Supplementary Figure 8).

Figure R5. EPR results of the reaction between BT and TPS at 453 K, with and without the addition of TEMPO as the radical scavenger.

5. As the authors mentioned that the ligand loading is not complete, it would be helpful to discuss possible strategies or future plans to improve ligand coverage, which could be important for further tuning of surface properties and device performance.

Response: We agree with the reviewer that the current thiophene ligand passivation is lower than typical aliphatic ligand passivation on SiNCs (REF: *Nanoscale*, **10**, 10337-10342, (2018); *Small* **15**, 1805400, (2019)). We attribute this primary to the steric hindrance from the bulky thiophene rings when directly anchor on SiNCs. As demonstrated in prior work with bulky ligands (REF: *Angew. Chem. Int. Ed.* **62**, e202304056, (2023)), such steric effects inherently limit the reactivity of neighboring Si-H sites in dehydrogenative coupling.

While conventional approaches such as kinetic optimization, we propose multi-step functionalization as a viable path to enable silicon-based nanoparticles being passivated with various types of ligands, and thus improve the surface ligand coverage (*Angew. Chem. Int. Ed.* **62**, e202304056, (2023); *Nanoscale*, **16**, 6516–6521, (2024)) are planning to implement this method to further functionalize the thiophene-capped SiNCs with aliphatic ligands such as hexane or octene to further improve surface coverage.

6. Regarding the ligand coverage, it would be helpful if the authors could provide a step-by-step explanation of the calculation method in SI.

Response: We thank the reviewer for the constructive suggestion. We have now updated the discussion of the surface ligand coverage calculation section (Supplementary Note 4) with step-by-step explanation about the detailed calculation process and the corresponding hypotheses.

7. In Fig. 3e, the nanocrystal diameter is reported as approximately 3 nm, whereas in the Supporting Information, the size is noted to be in the range of 2.3 – 2.5 nm. Please double check.

Response: We have reconfirmed that the size distributions of three SiNC samples reported in Supplementary Figure 14 are 2.38 ± 0.37 , 2.35 ± 0.33 , and 2.25 ± 0.34 nm. So, the average size of the reported SiNCs is about 2.4 ± 0.3 nm. We now follow the reviewer's suggestion, and change the description of the small particle diameter into 2.4 nm throughout the manuscript package.